# Association of the erythropoiesis-stimulating agent resistance index and the geriatric nutritional risk index with cardiovascular mortality in maintenance hemodialysis patients

**Takahiro Yajima**[1]*, **Kumiko Yajima**[2], **Hiroshi Takahashi**[3]

**1** Department of Nephrology, Matsunami General Hospital, Gifu, Japan, **2** Department of Internal Medicine, Matsunami General Hospital, Gifu, Japan, **3** Division of Medical Statistics, Fujita Health University School of Medicine, Aichi, Japan

* yajima5639@gmail.com

**Data Availability Statement:** All relevant data are within the paper and its Supporting Information files.

## Abstract

### Objective

Hyporesponsiveness to erythropoiesis-stimulating agent (ESA) may be associated with protein-energy wasting. We investigated the relationship of the ESA resistance index (ERI) and the geriatric nutritional risk index (GNRI) for cardiovascular mortality in hemodialysis (HD) patients.

### Methods

A total of 180 maintenance HD patients were enrolled. The patients were stratified by the GNRI of 91.2, a previously reported cut-off value, and the ERI of 13.7 (IU/week/kg/g/dL), a cut-off value for predicting cardiovascular-specific mortality, and they were classified into four groups (group 1[G1]: higher GNRI and lower ERI, G2: higher GNRI and higher ERI, G3: lower GNRI and lower ERI, G4: lower GNRI and higher ERI).

### Results

The ERI was independently associated with the GNRI (β = −0.271, p = 0.0005). During a median follow-up of 4.6 years, higher ERI and lower GNRI were independently associated with cardiovascular mortality, respectively (adjusted hazard ratio [aHR], 3.10; 95% confidence interval [CI], 1.31–7.34, and aHR, 6.64; 95%CI, 2.60–16.93, respectively). The 7-year survival rates were 96.1%, 70.3%, 77.3%, and 50.1% in G1, G2, G3, and G4, respectively. The aHR values for G4 versus G1 were 12.63 (95%CI, 3.58–44.59). With regards to model discrimination, adding the GNRI alone, the ERI alone, and both to the traditional risk model significantly improved the net reclassification improvement by 0.421, 0.662, and 0.671, respectively. Similar results were obtained for all-cause mortality.

**Funding:** The authors received no specific funding for this work.

**Competing interests:** The authors have declared that no competing interests exist.

## Conclusion

The ERI was independently associated with the GNRI, and could predict cardiovascular mortality in HD patients. Moreover, the combination of GNRI and ERI could improve the predictability for cardiovascular mortality.

## Introduction

Renal anemia, which is caused by decreased erythropoietin production due to kidney injury, is common among patients undergoing hemodialysis (HD), and is treated with erythropoiesis-stimulating agents (ESAs). It has been shown that HD patients who receive a high dose of ESAs relative to the hemoglobin (Hb) response experience poor outcomes, including increased risk of cardiovascular events or mortality[1–3]. It is not yet known whether these risks are caused by ESAs themselves, or underlying processes leading to increased ESA requirements. ESA hyporesponsiveness, or resistance, is generally defined as the requirement of higher than average doses of ESA to achieve an increase in Hb concentration[3–5]. The ESA resistance index (ERI) has been proposed as an indicator for ESA hyporesponsiveness, and some previous studies have shown that the ERI can predict all-cause mortality and/or cardiovascular events[6–8]. However, the associations between the ERI and cardiovascular mortality remain unclear.

The mechanisms of ESA hyporesponsiveness are not fully understood, but are likely to be multifactorial, relating to iron deficiency, inflammation, and malnutrition[9]. Recently, some studies have speculated that ESA hyporesponsiveness may be related to protein-energy wasting (PEW), a form of malnutrition characterized by loss of body protein and fuel reserves due to catabolic inflammation[10,11]. Okazaki et al. recently reported that high ERI and low geriatric nutritional risk index (GNRI) were associated with an increased risk of all-cause mortality in HD patients[12]. The GNRI can be used to classify patients according to a risk of complications in relation to conditions associated with PEW[13,14], and is also known to be an effective tool to identify those with malnutrition-related risks of all-cause and cardiovascular mortality in this population[15–17].

We investigated the associations of the ERI and the GNRI with cardiovascular and all-cause mortality in maintenance HD patients. In addition, we evaluated the combined predictability of the ERI and the GNRI for mortality in this population.

## Materials and methods

### Study participants

We conducted a retrospective cohort study of patients who had undergone maintenance hemodialysis therapy for at least 6 months. Patients who were treated with epoetin beta or darbepoetin alfa, but not with epoetin beta pegol for renal anemia were included. The study was performed using the medical records of the outpatient clinic of Matsunami General Hospital (Kasamatsu, Japan) between January 2008 and March 2020. Patients' data were fully anonymized prior to access, and as such, the requirement for informed consent was waived. This study adhered to the principles of the Declaration of Helsinki, and the study protocol was approved by the Ethics Committee of Matsunami General Hospital (No. 459).

### Data collection

The following patient data were collected from medical records: Age; sex; underlying kidney disease; duration of hemodialysis; history of alcohol, smoking, diabetes, hypertension, and

cardiovascular disease (CVD); dry weight; and height. CVD was defined as heart failure, angina pectoris, myocardial infarction, stroke, and peripheral artery disease. Diabetes was defined as a history or presence of diabetes, or prescription of glucose-lowering agents. Hypertension was defined as systolic blood pressure $\geq$ 140 mmHg and/or diastolic blood pressure $\geq$ 90 mmHg before hemodialysis, and/or prescription of anti-hypertensive drugs. Blood samples were collected in the supine position before hemodialysis sessions, which were conducted on either a Monday or a Tuesday. For the assessment of ESA responsiveness, the ERI was calculated by dividing the weekly weight-adjusted ESA dose (IU/week/kg) by the Hb concentration (g/dL)[6]. The darbepoetin alfa dose was harmonized with erythropoietin data by multiplying by 200[18,19]. The GNRI was calculated as follows: GNRI = (14.89 × albumin g/dL) + [41.7× (dry weight/ideal body weight)][13]. When the dry weight exceeded the ideal body weight, the "(dry weight/ideal body weight)" element was set to 1.

### Follow-up study

The primary endpoint was CVD mortality, and the secondary endpoint was all-cause mortality. Patients were divided by each cut-off point of ERI and GNRI; thereafter, patients were divided into four groups based on the combinations of each cut-off point of ERI and GNRI: Group 1 (G1), higher GNRI and lower ERI; G2, higher GNRI and higher ERI; G3, lower GNRI and lower ERI; and G4, lower GNRI and higher ERI. The patients were followed up until March 2020.

### Statistical analysis

Normally distributed variables are expressed as means ± standard deviations, and non-normally distributed variables are expressed as medians and interquartile ranges. The differences among the four subgroups divided by the GNRI and the ERI were compared by one-way analysis of variance or the Kruskal-Wallis test for continuous variables, or by the chi-squared test for categorical variables. Univariate regression analysis was performed to determine factors correlated with the ERI. Multivariate regression analysis was performed with the factors that were significantly associated with the ERI in the univariate analysis.

A cut-off value of GNRI 91.2 was used; this value was defined from a previous study[13]. Receiver operating characteristic (ROC) analysis was used to determine a cut-off value of the ERI to maximize the predictive value for cardiovascular-specific mortality. The Kaplan-Meier method was used to estimate the survival rate, and the difference was analyzed using the log-rank test. Hazard ratios (HRs) and 95% confidence intervals (CIs) for cardiovascular and all-cause mortality were calculated by Cox proportional hazard regression analysis. The multiple regression model included all covariates that were significant at $p < 0.05$ in the univariate analysis.

The C-index, net reclassification improvement (NRI), and integrated discrimination improvement (IDI) were calculated in order to assess whether the accuracy of predicting mortality improved after adding the GNRI and/or the ERI to the baseline model. The C-index was defined as the area under the receiver operating characteristic curve between individual predictive probabilities for mortality and the incidence of mortality. The C-index was compared between the baseline model, with all established risk factors, and the enriched model, including the GNRI and/or the ERI[20]. The NRI was used as a relative indicator of the number of patients for whom the predicted mortality risk improved, and the IDI was used to show the average improvement in predicted mortality risk after adding the new variables to the baseline model[21]. All statistical analyses were performed using IBM SPSS Version 21 (IBM Corp., Armonk, NY, USA). A p-value < 0.05 was considered statistically significant.

## Results

### Baseline characteristics

The baseline characteristics of the included patients are shown in Table 1. A total of 180 HD patients were included (age, 63.4 ± 13.9 years; male, 68.3%; HD duration, 0.6 [0.5–4.5] years; history of CVD, 67.8%). Hemoglobin, ferritin, transferrin saturation (TSAT), ESA dose, ERI, and GNRI levels were 10.7 ± 1.3 g/dL, 110 (49–201) ng/mL, 26.1 ± 12.7%, 4500 (4000–9000) IU/week, 8.7 (5.2–14.9) IU/week/kg/g/dL, and 94.5 ± 6.9, respectively. Univariate regression analysis showed that the ERI was significantly correlated with age ($\beta = 0.254$, p = 0.0006),

**Table 1. Baseline characteristics of the study patients.**

| | All patients (n = 180) | G1 (n = 93) | G2 (n = 20) | G3 (n = 32) | G4 (n = 35) | *p*-value |
|---|---|---|---|---|---|---|
| Age (years) | 63.4 ± 13.9 | 59.0 ± 13.6 | 67.2 ± 10.3 | 64.7 ± 16.3 | 71.5 ± 9.3 | < 0.0001 |
| Male (%) | 68.3 | 68.8 | 75.0 | 68.8 | 62.9 | 0.82 |
| Underlying kidney disease | | | | | | 0.92 |
| Diabetic kidney disease (%) | 45.0 | 48.4 | 45.0 | 37.5 | 42.9 | |
| Chronic glomerulonephritis (%) | 32.2 | 32.3 | 25.0 | 37.5 | 31.4 | |
| Nephrosclerosis (%) | 17.8 | 16.1 | 25.0 | 15.6 | 20.0 | |
| Others (%) | 5.0 | 3.2 | 5.0 | 9.4 | 5.7 | |
| HD duration (years) | 0.6 (0.5–4.5) | 1.0 (0.5–4.7) | 0.6 (0.5–3.8) | 0.5 (0.5–4.8) | 0.5 (0.5–3.9) | 0.55 |
| Alcohol (%) | 26.7 | 28.0 | 25.0 | 25.0 | 25.7 | 0.98 |
| Smoking (%) | 30.0 | 32.2 | 30.0 | 28.1 | 25.7 | 0.90 |
| Hypertension (%) | 95.6 | 96.8 | 100 | 90.6 | 94.3 | 0.30 |
| Diabetes (%) | 46.7 | 48.4 | 55.0 | 40.6 | 42.9 | 0.72 |
| History of CVD (%) | 67.8 | 66.7 | 60.0 | 65.6 | 77.1 | 0.54 |
| Dw (kg) | 57.7 ± 11.9 | 62.8 ± 11.9 | 52.2 ± 8.4 | 54.7 ± 8.2 | 50.2 ± 10.3 | < 0.0001 |
| BMI (kg/m$^2$) | 22.2 ± 3.6 | 23.8 ± 3.5 | 19.4 ± 2.9 | 22.0 ± 2.7 | 19.9 ± 2.9 | < 0.0001 |
| BUN (mg/dL) | 61.4 ± 16.7 | 64.9 ± 16.4 | 59.2 ± 13.2 | 58.0 ± 20.6 | 56.3 ± 14.0 | 0.028 |
| Creatinine (mg/dL) | 9.2 ± 3.2 | 9.8 ± 3.6 | 8.4 ± 2.2 | 9.0 ± 2.7 | 8.3 ± 2.1 | 0.050 |
| Single-pool Kt/V for urea | 1.3 ± 0.3 | 1.3 ± 0.3 | 1.4 ± 0.3 | 1.4 ± 0.3 | 1.4 ± 0.4 | 0.19 |
| Albumin (g/dL) | 3.7 ± 0.4 | 3.9 ± 0.3 | 3.4 ± 0.4 | 3.8 ± 0.2 | 3.3 ± 0.4 | < 0.0001 |
| Hemoglobin (g/dL) | 10.7 ± 1.3 | 11.0 ± 1.1 | 10.8 ± 1.2 | 10.1 ± 1.3 | 10.4 ± 1.5 | 0.0016 |
| T-Cho (mg/dL) | 152 ± 34 | 155 ± 35 | 146 ± 28 | 155 ± 34 | 142 ± 33 | 0.19 |
| Uric acid (mg/dL) | 7.1 ± 1.3 | 7.2 ± 1.7 | 7.4 ± 1.7 | 6.7 ± 2.3 | 6.8 ± 1.6 | 0.43 |
| Ca (mg/dL) | 8.8 ± 0.9 | 8.9 ± 0.9 | 8.5 ± 0.9 | 9.0 ± 0.8 | 8.7 ± 0.9 | 0.10 |
| P (mg/dL) | 5.2 ± 1.3 | 5.4 ± 1.3 | 4.7 ± 0.9 | 5.1 ± 1.5 | 4.9 ± 1.3 | 0.067 |
| iPTH (pg/mL) | 126 (45–215) | 142 (52–231) | 69 (23–130) | 115 (48–266) | 127 (25–210) | 0.064 |
| Glucose (mg/dL) | 138 ± 59 | 144 ± 66 | 141 ± 64 | 123 ± 39 | 133 ± 53 | 0.35 |
| CRP (mg/dL) | 0.15 (0.06–0.39) | 0.11 (0.06–0.28) | 0.24 (0.07–0.85) | 0.18 (0.06–0.33) | 0.29 (0.03–1.13) | 0.048 |
| Ferritin (ng/mL) | 110 (49–201) | 116 (60–204) | 144 (83–199) | 86 (27–167) | 99 (40–221) | 0.25 |
| TSAT (%) | 26.1 ± 12.7 | 27.3 ± 11.2 | 33.1 ± 20.6 | 23.1 ± 11.8 | 21.6 ± 9.1 | 0.0041 |
| ESA dose (IU/week) | 4500 (4000–9000) | 4500 (2250–4500) | 4000 (2250–4500) | 9000 (9000–9000) | 9000 (8000–9000) | < 0.0001 |
| ERI (IU/week/kg/g/dL) | 8.7 (5.2–14.9) | 6.1 (3.7–8.6) | 6.5 (4.8–7.5) | 16.8 (14.8–19.4) | 15.2 (11.8–19.9) | < 0.0001 |
| GNRI | 94.5 ± 6.9 | 98.8 ± 4.2 | 87.0 ± 3.8 | 96.1 ± 3.0 | 86.0 ± 5.0 | < 0.0001 |

*Abbreviations*: HD: Hemodialysis, BMI: Body mass index, BUN: Blood urea nitrogen, T-Cho: Total cholesterol, CRP: C-reactive protein, CVD: Cardiovascular disease, Dw: Dry weight, ESA: Erythropoiesis-stimulating agent, ERI: Erythropoiesis-stimulating agent resistance index, GNRI: Geriatric nutritional risk index, TSAT: Transferrin saturation

G1: Higher GNRI and lower ERI, G2: Higher GNRI and higher ERI, G3: Lower GNRI and lower ERI, G4: Lower GNRI and higher ERI.

**Table 2. Univariate and multivariate regression analysis of the associations between the erythropoiesis-stimulating agent resistance index and baseline variables.**

| Variables | Univariate | | Multivariate | |
|---|---|---|---|---|
| | β | *p*-value | β | *p*-value |
| Age | 0.254 | 0.0006 | 0.054 | 0.53 |
| Creatinine | -0.174 | 0.020 | -0.069 | 0.38 |
| TSAT | -0.326 | < 0.0001 | -0.289 | < 0.0001 |
| GNRI | -0.349 | < 0.0001 | -0.271 | 0.0005 |

*Abbreviations*: TSAT: Transferrin saturation, GNRI: Geriatric nutritional risk index.

creatinine (β = -0.174, p = 0.020), TSAT (β = -0.326, p <0.0001), and the GNRI (β = -0.349, p <0.0001). Multivariate regression analysis, following adjustment for all significant confounders in univariate analysis, revealed that the ERI was independently correlated with TSAT (β = -0.289, p <0.0001) and the GNRI (β = -0.271, p = 0.0005) (Table 2).

## Associations of the GNRI and ERI with CVD mortality

A total of 63 patients died during the follow-up period (4.6 [2.5–8.2] years), including 28 (44.4%) due to CVD-specific causes (14 heart failures, 7 sudden cardiac deaths or fatal arrhythmias, 4 strokes, and 3 myocardial infarctions), and 35 due to non-CVD-specific causes (25 infections, 7 malignancies, and 3 others).

In the multivariate Cox proportional hazards analysis adjusted by age, history of cardiovascular disease, creatinine, and C-reactive protein (CRP), which were significant in the

**Table 3. Cox proportional hazards analysis of the erythropoiesis-stimulating agent resistance index and the geriatric nutritional risk index for mortality.**

| Variables | Non-adjusted | | Adjusted* | |
|---|---|---|---|---|
| | HR (95%CI) | *p*-value | HR (95%CI) | *p*-value |
| Cardiovascular mortality | | | | |
| GNRI (continuous) | 0.87 (0.82–0.92) | < 0.0001 | 0.90 (0.84–0.96) | 0.0020 |
| ERI (continuous) | 1.09 (1.05–1.13) | < 0.0001 | 1.07 (1.02–1.11) | 0.0050 |
| Lower GNRI | 4.52 (2.10–9.73) | 0.0001 | 3.10 (1.31–7.34) | 0.0099 |
| Higher ERI | 8.19 (3.58–18.74) | < 0.0001 | 6.64 (2.60–16.93) | < 0.0001 |
| Cross-classified (vs. G1) | | < 0.0001 | | < 0.0001 |
| G2 | 6.77 (1.85–24.76) | 0.0039 | 6.70 (1.60–28.16) | 0.0094 |
| G3 | 8.62 (2.94–25.30) | < 0.0001 | 9.58 (2.83–32.45) | 0.0003 |
| G4 | 13.75 (4.74–39.88) | < 0.0001 | 12.63 (3.58–44.59) | < 0.0001 |
| All-cause mortality | | | | |
| GNRI (continuous) | 0.88 (0.85–0.91) | < 0.0001 | 0.92 (0.88–0.96) | 0.00012 |
| ERI (continuous) | 1.08 (1.05–1.10) | < 0.0001 | 1.06 (1.03–1.08) | 0.00019 |
| Lower GNRI | 5.14 (3.09–8.56) | < 0.0001 | 3.36 (1.92–5.87) | < 0.0001 |
| Higher ERI | 3.38 (2.01–5.70) | < 0.0001 | 2.49 (1.42–4.37) | 0.0015 |
| Cross-classified (vs. G1) | | < 0.0001 | | < 0.0001 |
| G2 | 6.05 (2.85–12.86) | < 0.0001 | 4.33 (1.93–9.72) | 0.0004 |
| G3 | 3.55 (1.70–7.44) | 0.0008 | 2.91 (1.34–6.32) | 0.0071 |
| G4 | 9.18 (4.73–17.82) | < 0.0001 | 5.87 (2.81–12.24) | < 0.0001 |

*Abbreviations*: ERI: Erythropoiesis-stimulating agent resistance index, GNRI: Geriatric nutritional risk index.

* Adjusted for age, history of cardiovascular disease, creatinine, and C-reactive protein, which were significant in the univariate analysis.

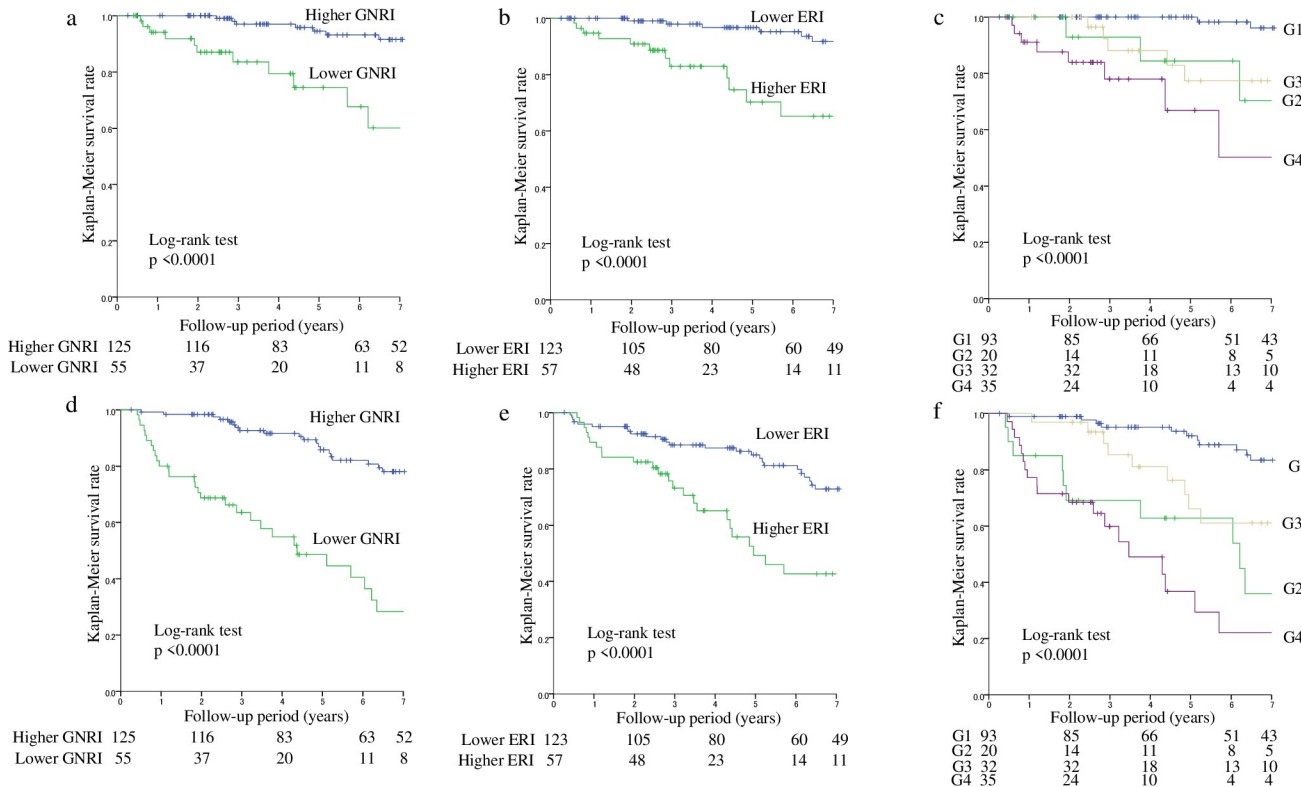

**Fig 1. Kaplan-Meier survival curves for cardiovascular and all-cause mortality.** Cardiovascular mortality for lower GNRI vs. higher GNRI (a), lower ERI vs. higher ERI (b), and among the four groups divided by the GNRI and the ERI (c). All-cause mortality for lower GNRI vs. higher GNRI (d), lower ERI vs. higher ERI (e), and among the four groups divided by the GNRI and the ERI (f). ERI: Erythropoiesis-stimulating agent resistance index, GNRI: Geriatric nutritional risk index. G1: Higher GNRI and lower ERI, G2: Higher GNRI and higher ERI, G3: Lower GNRI and lower ERI, G4: Lower GNRI and higher ERI.

univariate analysis, the GNRI and ERI were significant predictors for CVD mortality (HR, 0.87; 95%CI, 0.82–0.92, and HR, 1.09; 95%CI, 1.05–1.13, respectively) (Table 3). ROC analysis was performed to obtain the optimal cut-off values of the ERI for predicting the risk of CVD mortality. The cut-off value of the ERI was 13.7 IU/week/kg/g/dL (AUC = 0.655, p = 0.025). First, patients were divided by the GNRI of 91.2 into low and high groups, in which the 7-year CVD survival rates were 60.1% and 91.6%, respectively (p < 0.0001) (Fig 1A). Second, patients were then divided by the ERI of 13.7 IU/week/kg/g/dL into low and high groups, in which the 7-year CVD survival rates were 91.8% and 65.2%, respectively (p < 0.0001) (Fig 1B). Third, the patients were divided by each cut-off point of the GNRI and ERI into G1, G2, G3, and G4 groups, in which the 7-year CVD survival rates were 96.1%, 70.3%, 77.3%, and 50.1% (Fig 1C). Multivariate Cox proportional hazards analysis was performed after adjusting for age, history of CVD, creatinine, and CRP, which were significant in the univariate analysis. The adjusted HR (aHR) values for CVD mortality were 3.10 (95%CI, 1.31–7.34, p = 0.0099) for lower GNRI, and 6.64 (95%CI, 2.60–16.93, p < 0.0001) for higher ERI. Moreover, the aHR values were 6.70 (95%CI, 1.60–28.16, p = 0.0094) for G2 vs G1, 9.58 (95%CI, 2.83–32.45, p = 0.0003) for G3 vs G1, and 12.63 (95%CI, 3.58–44.59, p < 0.0001) for G4 vs G1 (Table 3). Similar results were obtained for all-cause mortality (Table 3, Fig 1D–1F).

With regards to the model discrimination, the C-index for CVD mortality was greater in the model adding the GNRI alone (0.708, p = 0.83), the ERI alone (0.747, p = 0.28), and both variables (0.753, p = 0.26) compared to the traditional risk model (0.708), but did not reach

**Table 4. Predictive accuracy of the erythropoiesis-stimulating agent resistance index and the geriatric nutritional risk index for mortality.**

| Variables | C-index | *p*-value | NRI | *p*-value | IDI | *p*-value |
|---|---|---|---|---|---|---|
| Cardiovascular mortality | | | | | | |
| Traditional risk factors* | 0.704 (0.589–0.820) | | Ref. | | Ref. | |
| + GNRI | 0.708 (0.589–0.827) | 0.83 | 0.421 | 0.020 | 0.011 | 0.093 |
| + ERI | 0.747 (0.650–0.844) | 0.28 | 0.662 | 0.00065 | 0.041 | 0.025 |
| + GNRI and ERI | 0.753 (0.659–0.846) | 0.26 | 0.671 | 0.00055 | 0.041 | 0.0097 |
| All-cause mortality | | | | | | |
| Traditional risk factors* | 0.722 (0.645–0.799) | | Ref. | | Ref. | |
| + GNRI | 0.744 (0.667–0.820) | 0.37 | 0.574 | 0.00012 | 0.051 | 0.0016 |
| + ERI | 0.729 (0.653–0.805) | 0.70 | 0.374 | 0.0084 | 0.021 | 0.043 |
| + GNRI and ERI | 0.767 (0.693–0.841) | 0.11 | 0.713 | <0.00001 | 0.072 | 0.0001 |

*Abbreviations*: ERI: Erythropoiesis-stimulating agent resistance index, GNRI: Geriatric nutritional risk index.

* Traditional risk factors include age, history of cardiovascular disease, creatinine, and C-reactive protein.

statistical significance. However, The NRI and IDI values for CVD mortality improved by adding the GNRI alone (0.421 [p = 0.020] and 0.011 [p = 0.093], respectively), the ERI alone (0.662 [p = 0.00065] and 0.041 [p = 0.025], respectively), and both variables (0.671 [p = 0.00055] and 0.041 [p = 0.0097], respectively) to the traditional risk model (Table 4). Similar results were obtained for all-cause mortality (Table 4).

## Discussion

The results of the present study showed that the ERI was negatively and independently associated with the GNRI, and could predict CVD and all-cause mortality in patients undergoing maintenance HD. Moreover, the combination of the ERI and the GNRI could not only stratify the risk, but also improve the predictability for mortality. Therefore, both the ERI and the GNRI should be evaluated to more accurately predict CVD mortality in this population.

The most common causes of ESA hyporesponsiveness are iron deficiency, either absolute or functional, inflammation, and malnutrition[9]. Absolute iron deficiency may be due to external blood losses through the extracorporeal blood circuit or dialyzers, and/or exhaustion of iron stores due to an increase in erythropoiesis caused by ESA treatment. In this situation, iron administration to maintain adequate iron stores is required for reducing the ESA dose and for enhancing ESA efficacy. However, some clinical trials have shown that iron administration to ESRD patients is associated with increased risks of infection, CVD, hospitalization, and mortality[22–24]. Functional iron deficiency is a condition in which iron utilization is defective in the bone marrow due to chronic inflammation despite sufficient iron stores. Malnutrition is closely related to inflammation and atherosclerosis[25], and through common mediators such as IL-6 or TNF-α, it may play a relevant role in ESA hyporesponsiveness[26].

On the other hand, PEW is a state of malnutrition, which is frequently complicated with chronic kidney disease, and is associated with an increased risk of mortality[27–29]. Some previous studies have reported that a loss of muscle mass and fat mass in the presence of inflammation leads to an increased risk of CVD mortality by promoting vascular endothelial damage [28–32]. As an indicator of PEW, the GNRI, a simple and objective method for evaluating nutritional status, is well-known in HD patients. Bouillanne et al. firstly reported that the GNRI was a prognostic indicator of morbidity and mortality in elderly hospitalized patients at nutritional risk[33]. Yamada et al. reported that the GNRI was the most reliable screening tool

for predicting malnutrition compared with other simple nutritional screening tools in maintenance hemodialysis patients[13]. They also determined the cutoff value of 91.2 for GNRI with the use of MIS as the standard reference, in this population[13]. Thereafter, many studies showed that the GNRI is a useful tool for stratifying malnutritional risks[14] and identifying nutrition-related risks of CVD events and all-cause or CVD mortality in HD patients [15,16,34]. A meta-analysis conducted by Xiong et al. also concluded that the GNRI is a significant indicator for predicting both all-cause and CVD mortality in patients undergoing HD [17]. Furthermore, we have recently reported that the ratio of extracellular fluid to intracellular fluid measured by bio-impedance analysis, a new marker of PEW, predicted not only all-cause, but also CVD mortality in patients undergoing HD[35]. Moreover, we have also revealed that the combining the ratio of extracellular fluid to intracellular fluid with the GNRI could improve predictability for mortality[35].

Several recent studies have reported the possible association between ESA hyporesponsiveness and PEW. Rattanasompattikul et al. have reported that the ERI was independently correlated with malnutritional-inflammation score, a comprehensive scoring system of nutrition in maintenance HD patients, and that the score was worse in the 4th quartile of ERI compared to the 1st quartile[10]. Furthermore, González-Ortiz et al. recently reported that HD patients with PEW, which was classified by malnutritional-inflammation score, have increased risks for the poorer response to ESA therapy than those without PEW[11]. In this study, the ERI was negatively, independently associated with GNRI, a marker of PEW; therefore, our findings supported that the ERI may be a plausible indicator of PEW.

In this study, higher ERI and lower GNRI were independently associated with CVD and all-cause mortality, respectively. Many observational studies have shown that the GNRI predicts all-cause and CVD mortality[15–17]. Although the associations between the ERI and all-cause mortality and CVD events has been already reported[6–8], the relationship between the ERI and CVD mortality remains unknown. Therefore, for the first time, we show that the ERI was a significant predictor for CVD mortality in HD patients. In the present study, the proportion of patients with a history of previous CVD was relatively high, and the study follow-up period was relatively long; thus, it was possible to determine the association between the ERI and CVD mortality. More interestingly, combining the ERI with the GNRI could stratify the risk of CVD mortality and improve the predictability. Therefore, both the ERI and the GNRI should be simultaneously evaluated in HD patients.

There were several limitations to this study. First, the present study was a single-center retrospective study with a relatively small number of participants. Second, patients with renal anemia who were treated with epoetin beta or darbepoetin alfa, but not with epoetin beta pegol were included. Since darbepoetin alfa but not epoetin beta pegol can be converted to the ESA dose of epoetin beta, our results might not be applicable to all patients in whom renal anemia is treated with ESAs. Third, the use of only baseline ERI and GNRI for data analysis was not allowed to consider any changes of these indicators during the follow-up periods. In addition, the changes of dialysis dose, nutritional status, and iron status markers might help to clarify the potential causes of a linked change of ERI, therefore future study may be needed to reveal these associations. Fourth, this study only included Japanese patients, and as such, our findings might not be representative of maintenance HD patients in other countries. Therefore, a further large-scale multicenter study may be needed to validate our results.

In conclusion, the ERI was independently associated with the GNRI and could predict CVD, as well as all-cause mortality in patients undergoing HD. Moreover, combining the ERI and the GNRI could not only stratify the risk of CVD and all-cause mortality, but could also improve the predictability for mortality. Therefore, both the ERI and the GNRI should be evaluated to more accurately predict CVD and all-cause mortality in this population.

## Supporting information

**S1 Data.**
(XLSX)

## Author Contributions

**Conceptualization:** Takahiro Yajima, Kumiko Yajima, Hiroshi Takahashi.

**Data curation:** Takahiro Yajima, Hiroshi Takahashi.

**Formal analysis:** Takahiro Yajima.

**Investigation:** Takahiro Yajima, Kumiko Yajima, Hiroshi Takahashi.

**Methodology:** Takahiro Yajima, Kumiko Yajima.

**Project administration:** Takahiro Yajima, Kumiko Yajima, Hiroshi Takahashi.

**Supervision:** Hiroshi Takahashi.

**Validation:** Takahiro Yajima, Hiroshi Takahashi.

**Writing – original draft:** Takahiro Yajima.

**Writing – review & editing:** Takahiro Yajima.

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
