## [Decision Letter · Decision Letter 0]

18 Dec 2020

PONE-D-20-33311

Association of the Erythropoiesis-Stimulating Agent Resistance Index and the Geriatric Nutritional Risk Index with Cardiovascular Mortality in Maintenance Hemodialysis Patients

PLOS ONE

Dear Dr. Takahiro Yajima,

Thank you for submitting your manuscript to PLOS ONE. After careful consideration, we feel that it has merit but does not fully meet PLOS ONE’s publication criteria as it currently stands. Therefore, we invite you to submit a revised version of the manuscript that addresses the points raised during the review .Please address comments by the reviewers.This topic has been already discussed in details in medical literature and will need more detailed submission in the discussion section to make it more impactful , this is a single center retrospective study on a topic already written lot in medical literature extensively .Some major  revisions are needed.

We look forward to receiving your revised manuscript.

Kind regards,

Bhagwan Dass, MD

Academic Editor

PLOS ONE

Additional Editor Comments:

Since you have mentioned that ERI was independently associated with the GNRI, and could predict cardiovascular mortality in HD patients. There are several existing studies showing relationship between the mechanisms of ESA hypo responsiveness and malnutrition. As we all know Malnutrition is closely related to inflammation and through mediators such as IL-6 or TNF- a, causing ESA hypo responsiveness, Both ESA hyporesposiveness and malnutrition based on current literature are closely related. The Geriatric Nutritional Risk Index (***GNRI***) is a simple screening tool to predict the risk of nutrition-related morbidity and mortality in mostly used in elderly patients. The population in your study group was age, 63.4 ± 13.9 years; this GNRI tool may not be uniformly applicable. The possible use of this tool GNRI.in HD patients may need some more studies to prove it as a more robust indicator of nutritional status in HD.patients.,Please add some more in the discussion section. .

2. In the ethics statement in the manuscript and in the online submission form, please provide additional information about the patient records/samples used in your retrospective study, including:  the date range (month and year) during which patients' medical records/samples were accessed.

3. Thank you for including your ethics statement:  "Patients’ data were fully anonymized prior to access, and as such, the requirement for informed consent was waived. This study adhered to the principles of the Declaration of Helsinki, and the study protocol was approved by the Ethics Committee (No. 459)".   

"No."

"No."

Reviewers' comments:

Reviewer's Responses to Questions

**Comments to the Author**

1. Is the manuscript technically sound, and do the data support the conclusions?

Reviewer #1: Yes

Reviewer #2: Yes

2. Has the statistical analysis been performed appropriately and rigorously? 

Reviewer #1: Yes

Reviewer #2: Yes

3. Have the authors made all data underlying the findings in their manuscript fully available?

Reviewer #1: Yes

Reviewer #2: Yes

4. Is the manuscript presented in an intelligible fashion and written in standard English?

Reviewer #1: Yes

Reviewer #2: Yes

5. Review Comments to the Author

Reviewer #1: Over all well done study, single center retrospective , evaluating ERI association with the GNRI to predict CVD, all-cause mortality in patients on hemodialysis. This study adds to our knowledge of risk stratification of patients on HD with ERI and GNRI, Few questions as below:

1. Please can you elaborate the concept that ERI can predict all-cause mortality and/or cardiovascular events but link with cardiovascular mortality remain unclear.

2. Table 1- HD duration is not clearly visible in different groups, can you please correct or its not showing correctly on the pdf. Is there a correlation with dialysis vintage to your finding?

Reviewer #2: Yajima T and coworkers have explored the predictive value of erythropoiesis-stimulating agent index (ERI) or geriatric nutritional risk index (GNRI) used alone versus used in combination (ERI plus GNRI) on all-cause and cardiovascular mortality in hemodialysis patients. For this purpose, they performed a retrospective cross-sectional study enrolling 180 prevalent maintenance HD patients. Patients were stratified according to the GNRI (threshold 91.2) and the ERI (threshold 13.7 IU/week/kg/g/dL). Four groups were then defined: group 1: higher GNRI and lower ERI, G2: higher GNRI and higher ERI, G3: lower GNRI and lower ERI, G4: lower GNRI and higher ERI. ERI was independently correlated with GNRI. Higher ERI and lower GNRI are independently associated with cardiovascular mortality. Survival rates are also inversely correlated with these predefined groups. It is also shown that ERI is independently associated with GNRI with a high predictive value for cardiovascular mortality. Furthermore, the combination of GNRI and ERI tend to improve cardiovascular mortality risk estimate.

This is an interesting study, well conducted and clearly written. However, this study is confirmatory by nature in showing that ERI is a sensitive and reliable marker of outcome in HD patients. It is originality relies in the additional use of GNRI either alone or in combination with ERI to assess mortality risk in HD patients.

My concerns are the following:

1. Looking at the composition of the GNRI equation two questions came in mind. Does serum albumin concentration alone would not have the same predictive value in assessing mortality risk in this population? Does ideal body weight relative to dry weight would not have the same predictive value in estimating mortality risk in this population? Therefore, what is the added value of using GNRI equation rather than albumin or body mass index as combined markers with ERI?

2. What is the role of inflammation marker (CRP) in this predictive estimate since it is already known as a strong marker and/or actor in this poor outcome complex? Interestingly, CRP seems higher in the group 4 but very low compared to European data.

3. What would be the additive value of simplified creatinine index as a more specific surrogate of protein energy wasting in the predictive value of ERI on dialysis patient outcomes?

4. What are the dynamic changes and predictive value of ERI and/or GNRI associated with these changes over time? That could be explored in this HD population since they were followed up to 7 years. It would be interesting to assess the fact that positive or negative trend changes in ERI or GNRI are associated with different outcomes.

5. Data summarizing dialysis efficiency would be of interest, as well as time trend changes of dialysis dose, nutritional, anemia and iron status markers to identify potential causes of high ERI or low GNRI.

6. PLOS authors have the option to publish the peer review history of their article (what does this mean?). If published, this will include your full peer review and any attached files.

Reviewer #1: No

Reviewer #2: No

---

## [Author Response · Author response to Decision Letter 0]

31 Dec 2020

Response to Reviewers

Response to editor

Thank you very much for your constructive comments.

Additional Editor Comments:

Since you have mentioned that ERI was independently associated with the GNRI, and could predict cardiovascular mortality in HD patients. There are several existing studies showing relationship between the mechanisms of ESA hypo responsiveness and malnutrition. As we all know Malnutrition is closely related to inflammation and through mediators such as IL-6 or TNF- a, causing ESA hypo responsiveness, Both ESA hyporesposiveness and malnutrition based on current literature are closely related. The Geriatric Nutritional Risk Index (GNRI) is a simple screening tool to predict the risk of nutrition-related morbidity and mortality in mostly used in elderly patients. The population in your study group was age, 63.4 ± 13.9 years; this GNRI tool may not be uniformly applicable. The possible use of this tool GNRI.in HD patients may need some more studies to prove it as a more robust indicator of nutritional status in HD patients. Please add some more in the discussion section. .

Thank you very much for the comments. According to your advice, we added more detailed explanations of GNRI in the third paragraph of discussion section as following: As an indicator of PEW, the GNRI, a simple and objective method for evaluating nutritional status, is well-known in HD patients. Bouillanne et al. firstly reported that the GNRI was a prognostic indicator of morbidity and mortality in elderly hospitalized patients at nutritional risk33. Yamada et al. reported that the GNRI was the most reliable screening tool for predicting malnutrition compared with other simple nutritional screening tools in maintenance hemodialysis patients13. They also determined the cutoff value of 91.2 for GNRI with the use of MIS as the standard reference, in this population13. Thereafter, many studies showed that the GNRI is a useful tool for stratifying malnutritional risks14 and identifying nutrition-related risks of CVD events and all-cause or CVD mortality in HD patients15,16,34. A meta-analysis conducted by Xiong et al. also concluded that the GNRI is a significant indicator for predicting both all-cause and CVD mortality in patients undergoing HD17. 

Response to Reviewer #1

Thank you very much for your constructive comments.

Reviewer #1: Over all well done study, single center retrospective , evaluating ERI association with the GNRI to predict CVD, all-cause mortality in patients on hemodialysis. This study adds to our knowledge of risk stratification of patients on HD with ERI and GNRI, Few questions as below:

1. Please can you elaborate the concept that ERI can predict all-cause mortality and/or cardiovascular events but link with cardiovascular mortality remain unclear.

Thank you very much for the comment. 

As mentioned in the Introduction section, some previous studies showed that the ERI was a predictor for all-cause mortality and cardiovascular events. If the rates of cardiovascular events increase, those of cardiovascular mortality might increase. However, there is no study which investigated the associations between the ERI and cardiovascular mortality. Thus, we investigated these associations in the present study. 

2. Table 1- HD duration is not clearly visible in different groups, can you please correct or its not showing correctly on the pdf. Is there a correlation with dialysis vintage to your finding?

Thank you very much for the comments. 

We showed that HD duration is clearly visible in the divided four groups on the PDF. HD vintage was not a significant predictor in the univariate analysis (HR 1.029, 95%CI 0.955-1.091). Moreover, even after the addition of HD vintage into multivariate model, higher ERI was still independently associated with an increased risk of CVD mortality (HR 6.017, 95%CI 2.381-15.21). 

Response to Reviewer #2

Thank you very much for your constructive comments.

Reviewer #2: Yajima T and coworkers have explored the predictive value of erythropoiesis-stimulating agent index (ERI) or geriatric nutritional risk index (GNRI) used alone versus used in combination (ERI plus GNRI) on all-cause and cardiovascular mortality in hemodialysis patients. For this purpose, they performed a retrospective cross-sectional study enrolling 180 prevalent maintenance HD patients. Patients were stratified according to the GNRI (threshold 91.2) and the ERI (threshold 13.7 IU/week/kg/g/dL). Four groups were then defined: group 1: higher GNRI and lower ERI, G2: higher GNRI and higher ERI, G3: lower GNRI and lower ERI, G4: lower GNRI and higher ERI. ERI was independently correlated with GNRI. Higher ERI and lower GNRI are independently associated with cardiovascular mortality. Survival rates are also inversely correlated with these predefined groups. It is also shown that ERI is independently associated with GNRI with a high predictive value for cardiovascular mortality. Furthermore, the combination of GNRI and ERI tend to improve cardiovascular mortality risk estimate.

This is an interesting study, well conducted and clearly written. However, this study is confirmatory by nature in showing that ERI is a sensitive and reliable marker of outcome in HD patients. It is originality relies in the additional use of GNRI either alone or in combination with ERI to assess mortality risk in HD patients.

My concerns are the following:

1. Looking at the composition of the GNRI equation two questions came in mind. Does serum albumin concentration alone would not have the same predictive value in assessing mortality risk in this population? Does ideal body weight relative to dry weight would not have the same predictive value in estimating mortality risk in this population? Therefore, what is the added value of using GNRI equation rather than albumin or body mass index as combined markers with ERI?

Thank you very much for the comments. 

As you mentioned above, the GNRI is considered to be a marker composed with albumin (Alb) and body mass index (BMI). In the present study, the increased of Alb (HR 0.115, 95%CI 0.047-0.295, p <0.0001) and BMI (HR 0.886, 95%CI 0.781-0.993, p = 0.037) were also associated with decreased risks of cardiovascular mortality, respectively as well as the GNRI. However, Takahashi et al. (Journal of Cardiology 2014;64:32–36) have already reported that the predictability of all-cause and cardiovascular mortality improved when GNRI was added into baseline model compared to when Alb or BMI was added. Therefore, we think the GNRI rather than Alb or BMI may be suitable as a combined marker with ERI. We thanks for your kind understanding.

2. What is the role of inflammation marker (CRP) in this predictive estimate since it is already known as a strong marker and/or actor in this poor outcome complex? 

Thank you very much for the comments.

As reviewer mentioned, CRP was a significant predictor for CVD mortality (HR 1.486, 95%CI 1.152-1.807, p=0.0054) by univariate analysis, therefore we included it in multivariate Cox proportional hazards analysis, and the results were shown in table 3.

Interestingly, CRP seems higher in the group 4 but very low compared to European data.

Thank you very much for the comment. 

CRP concentrations are five times lower in Japan than in Europe (Lancet. 2016; 388: 294–306). Thus, we think our results were not surprising. 

3. What would be the additive value of simplified creatinine index as a more specific surrogate of protein energy wasting in the predictive value of ERI on dialysis patient outcomes?

Thank you very much for the comment. 

As you mentioned above, simplified creatinine index has been recently developed as a surrogate marker of lean body mass and is calculated using age, sex, pre-dialysis serum creatinine level, and single-pool Kt/V for urea. In the present study, simplified creatinine index was 20.4 ± 3.1 mg/kg/d. In univariate Cox proportional hazards analysis, simplified creatinine index was a significant predictor of cardiovascular mortality (HR 0.741, 95%CI 0.643-0.846, p <0.0001). After adjusting for age, previous history of CVD, and CRP, simplified creatinine index was an independent predictor of cardiovascular mortality (HR 0.825, 95%CI 0.692-0.984, p = 0.033).

However, the modified creatinine index is affected by day-to-day dietary protein intake and dialysis dose, and it may be affected by residual kidney function (J Ren Nutr. 2020 Sep 17:S1051-2276(20)30211-9, Am J Kidney Dis. 2020 Feb;75(2):195-203.). Because the median HD vintage of the present study participants was 0.6 years, therefore the influence of residual kidney function to modified creatinine index might not be ignored. Moreover, Yamada et al. reported that GNRI and modified creatinine index equally predicted the risks of mortality in patients undergoing HD (scientific reports 2020;10:5756). Thus, we think it may be appropriate to use GNRI instead of modified creatinine index for evaluating the predictability of the ERI in the present study.

4. What are the dynamic changes and predictive value of ERI and/or GNRI associated with these changes over time? That could be explored in this HD population since they were followed up to 7 years. It would be interesting to assess the fact that positive or negative trend changes in ERI or GNRI are associated with different outcomes.

Thank you very much for the comments. We fully agree to your comments. 

Unfortunately, we have only baseline ERI and GNRI data, which was approved by the Ethics Committee, in the present study. However, we think that this is a very important issue, therefore we state it in the limitation. 

5. Data summarizing dialysis efficiency would be of interest, as well as time trend changes of dialysis dose, nutritional, anemia and iron status markers to identify potential causes of high ERI or low GNRI.

Thank you very much for the comments. We deeply agree to your comments. 

We added single-pool Kt/V for urea as a baseline dialysis dose in the table 1. The baseline ERI was not associated with baseline single-pool Kt/V for urea (β = 0.099, p = 0.18). However, as we mentioned above, we have no follow-up data, therefore we cannot investigate the relationships between ERI or GNRI and the changes of dialysis dose, nutritional, and anemia and iron status markers. We think that this is a limitation, therefore we also state it in our manuscript.

---

## [Editor Report · Decision Letter 1]

5 Jan 2021

Association of the Erythropoiesis-Stimulating Agent Resistance Index and the Geriatric Nutritional Risk Index with Cardiovascular Mortality in Maintenance Hemodialysis Patients

PONE-D-20-33311R1

Dear Dr. Yajima,

We’re pleased to inform you that your manuscript has been judged scientifically suitable for publication and will be formally accepted for publication once it meets all outstanding technical requirements.

Kind regards,

Bhagwan Dass, MD

Academic Editor

PLOS ONE
---

## [Editor Report · Acceptance letter]

7 Jan 2021

PONE-D-20-33311R1 

Association of the Erythropoiesis-Stimulating Agent Resistance Index and the Geriatric Nutritional Risk Index with Cardiovascular Mortality in Maintenance Hemodialysis Patients 

Dear Dr. Yajima:

I'm pleased to inform you that your manuscript has been deemed suitable for publication in PLOS ONE. Congratulations! Your manuscript is now with our production department. 

Kind regards, 

on behalf of

Dr. Bhagwan Dass 

Academic Editor

PLOS ONE